# ENVIRONMENT PROBING INTERACTION POLICIES

**Wenxuan Zhou[1], Lerrel Pinto[1], Abhinav Gupta[1,2]**
[1]The Robotics Institute, Carnegie Mellon University
[2]Facebook AI Research

## ABSTRACT

A key challenge in reinforcement learning (RL) is environment generalization: a policy trained to solve a task in one environment often fails to solve the same task in a slightly different test environment. A common approach to improve inter-environment transfer is to learn policies that are invariant to the distribution of testing environments. However, we argue that instead of being invariant, the policy should identify the specific nuances of an environment and exploit them to achieve better performance. In this work, we propose the "Environment-Probing" Interaction (EPI) policy, a policy that probes a new environment to extract an implicit understanding of that environment's behavior. Once this environment-specific information is obtained, it is used as an additional input to a task-specific policy that can now perform environment-conditioned actions to solve a task. To learn these EPI-policies, we present a reward function based on transition predictability. Specifically, a higher reward is given if the trajectory generated by the EPI-policy can be used to better predict transitions. We experimentally show that EPI-conditioned task-specific policies significantly outperform commonly used policy generalization methods on novel testing environments.

## 1 INTRODUCTION

Over the last few years deep reinforcement learning (RL) has shown tremendous progress. From beating the world's best at Go (Silver et al., 2016) to generating agile and dexterous motor policies (Schulman et al., 2015; Lillicrap et al., 2015; Peng et al., 2018), deep RL has been used to solve a variety of sequential decision-making tasks. However, one key issue that deep RL faces is transfer: a policy trained in one environment is extremely dependent on the specific parameters of that environment (Rajeswaran et al., 2017) and fails to generalize to a new environment. Because of this, training agents that can generalize to novel environments has become an active area of research (Rajeswaran et al., 2017; Pinto et al., 2017b; Yu et al., 2017; Peng et al., 2017).

The most common approach for environment generalization is to train a single policy over a variety of different environments (Rajeswaran et al., 2017; Peng et al., 2017), thus allowing the policy network to learn invariances to the underlying environmental factors. However, a monolithic policy that works on multiple environments often ends up being conservative since it is unable to exploit the dynamics of a given environment. We argue that instead of learning a policy that is invariant to environment dynamics we need a policy that adapts to its environment. A popular approach for this is to first perform explicit system identification of the environment, and then use the estimated parameters to generate an environment specific policy (Yu et al., 2017). However, there are three reasons why this approach might be infeasible: (a) it is based on the structural assumption that only a known set of parameters matter; (b) estimating those parameters from few observations is an extremely challenging problem; (c) these approaches require access to environment parameters during training time, which might be impractical. So is it possible to learn an environment dependent policy without estimating every environment parameter?

When humans are tasked to perform in a new environment, we do not explicitly know what parameters affect performance (Braun et al., 2009). Instead, we probe the environment to gain an intuitive understanding of its behavior. The purpose of these initial interactions is not to complete the task immediately, but to extract information about the environment. This process facilitates learning in that environment. Inspired by this, we present a framework in which the agent first performs "environment-probing" interactions that extract information from an environment, then leverages

this information to achieve the goal with a task-specific policy. This setup entails three challenges: 1) How should an agent interact with an environment? 2) How can information be extracted from such interactions? 3) How can the extracted information be used to learn a better policy?

One way to interact is by executing random actions in the environment. However, this may not expose the nuances of an environment. Another way is to use samples from the task-specific policy to infer the environment (Yu et al., 2017). However, a good task-specific policy may not be optimal for understanding an environment. Instead, we separately optimize an "Environment-Probing" Interaction policy (EPI-policy) to extract environment information. To assign rewards to the EPI-policy, our key insight is that the agent's ability to predict transitions in an environment can be used as a metric of its understanding of that environment. If an "environment-probing" trajectory can help in predicting transitions, then it should be a good trajectory. More specifically, we use the "environment-probing" trajectory to generate an environment embedding vector. This compact embedding vector is passed to a prediction model that provides higher rewards to the trajectories that are useful in estimating environment-dependent transitions.

Once we have learned the EPI-policy, we can use the generated environment embedding as an input to the task-specific policy. Since this task-specific policy is conditioned on the information from the environment, it can now produce actions dependent on the nuances of the specific environment.

We evaluate the effectiveness of using EPI on Hopper and Striker from OpenAI Gym (Brockman et al., 2016) with randomized mass, damping and friction. In both of these environments, the performance of our policies is better than several commonly used techniques and is even comparable to an oracle policy that has full access to the environment information. These results, along with a detailed ablation analysis, demonstrates that implicitly encoding the environment using EPI-policies improves task performance in unseen testing environments.

## 2 RELATED WORK

Developing generalizable machine learning models is a longstanding challenge. One approach to transfer models across different input distributions is domain adaptation. Duan et al. (2012) along with several others (Hoffman et al., 2014; Kulis et al., 2011; Long et al., 2015) domain-adapt visual models. With the onset of deep learning, several works have investigated finetuning (retraining) pre-trained networks in different target domains (Yosinski et al., 2014; Li et al., 2016; Tzeng et al., 2014). In reinforcement learning, Gupta et al. (2017) learns different encoders for different environments that share a common feature space to enable transfer. Improving the generalizability of models can also be formulated as a meta-learning problem. Finn et al. (2017) and Yu et al. (2018) finetune the models in new environments. We argue that agents should have the capacity to generalize without updating their policy. Duan et al. (2016b) and Wang et al. (2016) optimize the policy over multiple episodes and use RNN to implicitly model the underlying changing dynamics in the hidden states. Hausman et al. (2018) purposes to learn an embedding in a supervised manner by performing variational inference on task ids. These methods are based on model-free meta-RL, while Sæmundsson et al. (2018) and Clavera et al. (2018) investigate in model-based meta-RL. Sæmundsson et al. (2018) infers latent variables using Gaussian Process, while we use neural networks to create an embedding and experimented with more complex high-dimensional continuous control problems. Concurrent work Clavera et al. (2018) proposes to train a dynamics model that adapts online using either gradient based method (Finn et al., 2017) or recurrent based method (Duan et al., 2016b). The most important difference is that we have a separate policy to probe the environment instead of using the trajectories from the execution of the task. In our method, training prediction models is somewhat related to learning dynamics model in the model-based RL. However, the prediction model in our method does not to be very accurate since we are only taking the difference.

Another stream of research that is relevant to our work is simulation-to-real transfer. Here the goal is to train a policy in a physics simulator and then transfer that policy to a real robot. The challenge here is to bridge the "Reality Gap" between the real world and the simulator. Rusu et al. (2016) attempt to do this via architectural novelty for visual domain adaptation. Sadeghi & Levine (2016); Tobin et al. (2017); Pinto et al. (2017a) and James et al. (2017) perform domain randomization on the inputs during training to learn a transferable policy. These methods target the mismatch of observation space. However, we target the mismatch in the dynamics of the environment. Peng et al. (2017) proposes domain randomization of the environment parameters and learning an LSTM policy to

implicitly learn the environment. We compare our method to this approach and show that a separate process that extracts the nuances of an environment is vital for better generalization. Rajeswaran et al. (2017) and Pinto et al. (2017b) also target generalization to environments, however they do not incorporate environment information and hence are only able to learn a conservative policy.

The traditional approach of system identification (Milanese et al., 2013; Gevers et al., 2006; Bongard & Lipson, 2005; Armstrong, 1987) aims to explicitly estimate the varying environment parameters. Yu et al. (2017) propose the UP-OSI framework that can perform online system identification and takes the estimated environment parameters as additional input to the policy. This works well when the number of parameters to estimate is few. However, explicit identification of every environment parameter is a extremely challenging. Denil et al. (2016) trains an policy to gather information about physical properties through interactions in intuitive physics to answer questions, while we are trying do dealt with a more complex continuous control setting.

A recent approach by Pathak et al. (2017; 2018) uses errors in prediction models as an intrinsic reward (Chentanez et al., 2005) to improve exploration in RL. The strategy is to take actions to reach the states that maximize the error in the learned prediction model. In contrast, our method selects actions that reduce the prediction error on the heldout prediction dataset of task policy trajectories.

## 3    BACKGROUND: REINFORCEMENT LEARNING

We use the framework of reinforcement learning to learn the interaction policy and the final task target policy. This section contains brief preliminaries for reinforcement learning. Note that a more comprehensive exposition can be found in Kaelbling et al. (1996) and Sutton & Barto (1998).

We model our problem as a continuous space Markov Decision Process represented as the tuple $(\mathcal{S}, \mathcal{A}, \mathcal{P}, r, \gamma, \mathbb{S})$, where $\mathcal{S}$ is a set of continuous states of the agent, $\mathcal{A}$ is a set of continuous actions the agent can take at a given state, $\mathcal{P} : \mathcal{S} \times \mathcal{A} \times \mathcal{S} \rightarrow \mathbb{R}$ is the transition probability function, $r : \mathcal{S} \times \mathcal{A} \rightarrow \mathbb{R}$ is the reward function, $\gamma$ is the discount factor, and $\mathbb{S}$ is the initial state distribution.

The goal for an agent is to maximize the expected cumulative discounted reward $\sum_{t=0}^{T-1} \gamma^t r(s_t, a_t)$ by performing actions according to its policy $\pi_\theta : \mathcal{S} \rightarrow \mathcal{A}$. Here, $\theta$ denotes the parameters for the policy $\pi$ which takes action $a_t$ given state $s_t$ at timestep $t$. There are various techniques to optimize $\pi_\theta$. The one we use in this paper is a policy gradient approach, TRPO (Schulman et al., 2015).

## 4    APPROACH

Our objective is to learn a generalizable policy that is trained on a set of training environments, but can succeed across unseen testing environments. To this end, instead of a single policy, we have two separate policies: (a) an Environment-Probing Interaction policy (EPI-policy) that efficiently interacts with an environment to collect environment-specific information, and (b) a task-specific policy that uses this additional environment information (estimated via EPI-policy) to achieve high rewards for a given task. In this section, we present our methodology for learning these two policies.

### 4.1    LEARNING ENVIRONMENT-PROBING INTERACTIONS (EPI)

To learn the EPI-policy, we introduce a reward formulation based on the prediction models (Fig. 1). The key insight here is that a good EPI-policy generates trajectories in an environment from which predicting transitions in that environment is easier. To quantify this notion of how easy or difficult it is to predict transitions, we use the error transition prediction models.

### 4.1.1    TRANSITION PREDICTION MODELS

To train our prediction models, a dataset of transition data $(s_t, a_t, s_{t+1})$ is collected in the training environments using a pre-trained task policy (Sec. 4.1.3). This data is split into a training set and a validation set. With the training set, we train two prediction models: (a) a vanilla prediction model $f(s_t, a_t)$ that predicts the next state $s_{t+1}$ given the current state $s_t$ and action $a_t$, and (b) an EPI-conditioned prediction model $f_{epi}(s_t, a_t; \psi(\tau_{epi}))$, which takes the embedded EPI-trajectory $\psi(\tau_{epi})$ as an additional input. Here $\tau_{epi}$ is a trajectory in the environment which contains a small

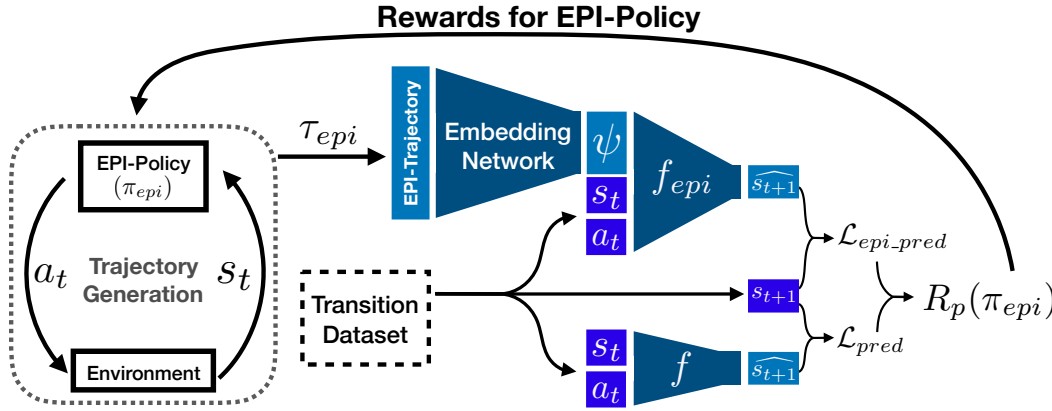

Figure 1: We illustrate the architecture for learning the EPI-Policy. Trajectories $\tau_{epi}$ generated from the EPI policy are passed through an embedding network to obtain the environment embeddings $\psi$. The difference in performance between a simple prediction model $f$ and the embedding-conditioned prediction model $f_{epi}$ is used to train the EPI-policy.

fixed number of observations and actions generated by $\pi_{epi}$. $\psi(\tau_{epi})$ is the low dimensional embedding of the trajectory $\tau_{epi}$. We can now quantify the reward for $\pi_{epi}$ as the difference between their performance. The intuition behind this is that if an interaction trajectory generated from $\pi_{epi}$ is informative, $f_{epi}$ will be able to use it to give smaller prediction error than $f$.

To train the prediction models, we use a mean squared error loss, i.e the loss for $f$ is:

$$\mathcal{L}_{pred} = \frac{1}{N} \sum_{i=1}^{N} \|f(s_t, a_t) - s_{t+1}\|_2^2 \tag{1}$$

Here $s_{t+1}$ is the target next state and $\|\cdot\|_2$ is the L2 norm. Similarly, the loss for $f_{epi}$ is:

$$\mathcal{L}_{epi\_pred} = \frac{1}{N} \sum_{i=1}^{N} \|f(s_t, a_t; \psi(\tau_{epi})) - s_{t+1}\|_2^2 \tag{2}$$

During training, gradients are back-propagated to train both the prediction model $f_{epi}$ and the embedding network $\psi$. The embedding network $\psi$ will be then directly used for the task-specific policy.

### 4.1.2 EPI REWARD FORMULATION

The reward for the EPI-policy is computed as the difference in performance between the simple prediction model $f$ and the EPI dependent prediction model $f_{epi}$ over the validation transition dataset. For a given training environment, the trajectory $\tau_{epi}$ generated by EPI-policy $\pi_{epi}$ is fed into $f_{epi}$ to calculate the prediction loss $\mathcal{L}_{epi\_pred}$. The simple prediction loss $\mathcal{L}_{pred}$ is calculated using $f$. This leads us to the prediction reward function:

$$R_p(\pi_{epi}) = \mathcal{L}_{pred} - \mathbb{E}_{\tau_{epi} \sim \pi_{epi}}[\mathcal{L}_{epi\_pred}(\tau_{epi})] \tag{3}$$

Note that this formulation is different from the "curiosity" reward proposed by Pathak et al. (2017) which encourages the policy to visit unexplored regions of the state space. Our reward function is to encourage the policy to extract information to improve prediction, rather than explore.

### 4.1.3 ADDITIONAL TRAINING DETAILS

**Collecting transition dataset:** To collect the transition dataset, we first train a task policy over a fixed environment. This policy is then executed with an $\epsilon$-greedy strategy over the training environments to generate the transition data. To make this dataset more environment-dependent, we collect more data by setting different environments to the same state and executing the same action, following the Vine method (Schulman et al., 2015). Different environments would cause different observations in the next timestep, thus forcing the prediction model to use environment information given by the interaction. The dataset is kept fixed throughout the updates of the EPI-policy.

**Regularization via Separation:** An additional regularization can be added to the loss function of the EPI-conditioned prediction model $f_{epi}$. We implement a separation loss back-propagated from the outputs of the embedding network $\psi(\tau_{epi})$. The idea is to additionally force the embeddings from different environments to be apart from each other when the interaction trajectories are not yet informative enough for prediction. To calculate this loss, we assume the knowledge of which trajectory is from the same environment. We then fit a multivariate Gaussian distribution over the embeddings in an environment $\mathcal{E}_i$ and encourage the mean vector $\mu_i$ of one environment to be at least a fixed distance away from every other environment $\mu_{j \neq i}$. In addition, we encourage the diagonal elements of the covariance matrix to be reasonably small.

**Interleaved training:** Since the EPI-policy and the prediction model depend on each other, they are optimized in an alternating fashion. The prediction model is first trained using a randomly initialized EPI-policy. Then the EPI-policy is optimized using TRPO for a few iterations with rewards from the learned prediction model. After that, the prediction model is updated again using a batch of trajectories generated from the EPI-policy. This cycle of alternating optimization is repeated until the EPI-policy converges. Details on the exact parameters used can be found in Appendix C.

### 4.2 Learning task-specific policies

After the EPI-policy and the embedding network are optimized, they are fixed and used for training the task-specific policy. For each episode, the agent first follows the EPI-policy to generate a trajectory $\tau_I$. The trajectory is then converted to an environment embedding $\psi(\tau_I)$ using the trained embedding network $\psi$ from the prediction model. After the interaction stage, the agent starts an episode to achieve its goal. This environment embedding $\psi(\tau_I)$ is now a part of the observation for each time step of the task-specific policy $\pi_{task}(s_t, \psi(\tau_I))$. The task-specific policy is then optimized using TRPO. During testing, the agent follows the same procedure: it first executes the EPI-policy to get the environment embedding, followed by the task-specific policy to achieve the goal.

## 5 Results

We now present experimental results to demonstrate how the EPI-policy helps the agent understand the environment, and allows the task-specific policy to generalize better to unseen test environments. Code is available at https://github.com/Wenxuan-Zhou/EPI.

**Environments:** To evaluate the EPI-policy, we need appropriate simulation environments that are sensitive to environment parameters. For this, we use the Striker and the Hopper MuJoCo (Todorov et al., 2012) environments from OpenAI Gym (Brockman et al., 2016). Both of the agents are torque-controlled. For Striker, the goal is to strike an object on the table with a robot arm to make the object reach at a target position when it stops. The mass and damping of the object are varied (2 parameters). The goal of Hopper is to move as fast as possible in the forward direction without falling. Here, we vary the mass of four body parts, the damping of three joints and the friction coefficient of the ground (8 parameters). During training, each parameter is randomized uniformly among five values from its range at the beginning of an episode. During testing, the parameters are sampled from a set of unseen environment parameters. Details can be found in Appendix A and Appendix B.

**Baselines:** To show that the EPI-policy improves generalization, we compare our results with three categories of baselines. Training details for the baselines can be found in Appendix D. First, we show the importance of extra environment information:

- Simple Policy: a MLP (Multi-Layer Perceptron) policy $a_t = \pi(s_t)$ trained only in one fixed environment of fixed parameters. This is how RL tasks are normally formulated.

- Invariant Policy (*inv-pol*): a MLP policy $a_t = \pi(s_t)$ trained across the randomized environments similar to Rajeswaran et al. (2017). This policy will be more robust than Simple Policy since it encounters various environments during training. However, it is only able to output the same action from the same input in different environments.

- Oracle Policy: a MLP policy takes explicit environment parameters as additional input: $a_t = \pi(s_t, \rho)$, where $\rho$ is a vector of the environment parameters such as mass, friction and damping. This baseline has full information of the environment, thus this is meant to put our results in context with the best possible task-specific policy.

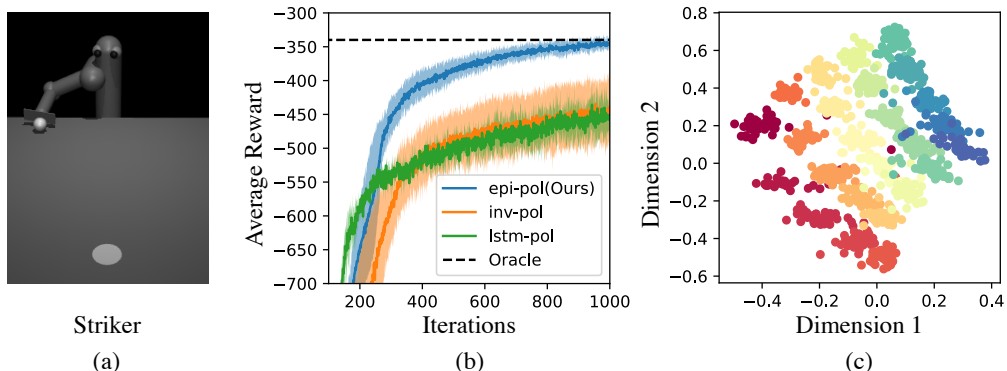

Figure 2: Striker Environment: (a) Illustration of the environment (b) Training curves of the EPI-policy comparing to baselines. The x-axis shows training iterations for TRPO. The y-axis shows the average reward of an episode.(c) Two-dimensional environment embedding of the EPI-policy after training. Embeddings from the same environment have the same color.

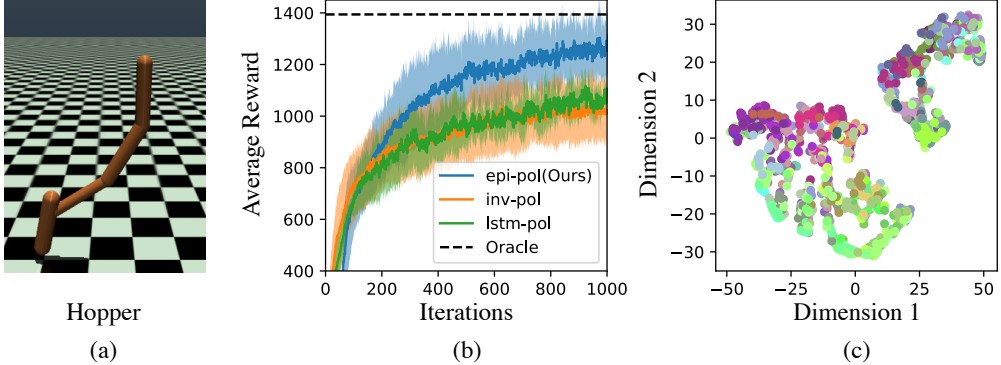

Figure 3: Hopper Environment: (a) Illustration of the environment (b) Training curves of the EPI-policy comparing to baselines. The x-axis shows training iterations. The y-axis shows the average reward of an episode. (c) t-SNE of the 8-dimensional Environment embedding of the EPI-policy after training. To visualize, the color is given by (R,G,B) where R:average of mass, G:friction, B:average of damping.

In the second category of baselines, we want to demonstrate the necessity of learning a separate interaction policy:

- Random Interaction Policy: takes 10 timesteps of random interactions as additional input. It covers the possibility that random actions are good enough to capture environment nuances.

- History Policy: takes 10 timesteps of most recent states and actions as additional input. Mnih et al. (2015) adopts similar idea where it stacks 3 to 5 most recent frames as input. It covers the possibility that the trajectories from the task-specific policy are informative.

- Recurrent Policy (*lstm-pol*): using a LSTM layer instead of MLP. This baseline is similar to Peng et al. (2017). LSTM is supposed to encode the past trajectories, so it serves the same idea as the History Policy but from a different approach.

- System Identified Policy: This baseline is implemented based on Yu et al. (2017). It first trains an Oracle Policy. Then it performs explicit system identification that estimates environment parameters using a history of states and actions from the Oracle Policy. Finally, the estimated value are fed into the trained Oracle Policy during test time.

Finally, to show that the prediction reward is an appropriate proxy for the EPI-policy update, we implemented the following baseline that directly updates the EPI-policy:

- Direct Reward Policy: this baseline evaluates the interactions by directly taking the final reward of the following task-specific policy trajectory instead of the prediction reward.

Table 1: Comparison of testing results. Hopper is evaluated by the accumulated reward in one episode. Higher reward implies better performance. Striker is evaluated by the final distance between the object and the goal. A smaller distance implies a better performance.

| METHOD | | Hopper: Reward ($\uparrow$) | Striker: Final Distance ($\downarrow$) |
|---|---|---|---|
| | Simple Policy | $414 \pm 313$ | $1.660 \pm 2.010$ |
| | Invariant Policy | $1025 \pm 49$ | $0.297 \pm 0.068$ |
| | Random Interaction Policy | $1101 \pm 27$ | $0.410 \pm 0.047$ |
| Baselines | History Policy | $1143 \pm 156$ | $0.259 \pm 0.038$ |
| | Recurrent Policy | $917 \pm 180$ | $0.418 \pm 0.051$ |
| | System Id Policy | $1033 \pm 81$ | $1.113 \pm 0.106$ |
| | Direct Reward | $1057 \pm 310$ | $0.458 \pm 0.004$ |
| **Ours** | **EPI + Task-specific Policy** | $\mathbf{1303 \pm 173}$ | $\mathbf{0.162 \pm 0.015}$ |
| | No Vine Data | $1214 \pm 138$ | $0.293 \pm 0.018$ |
| Ablations | No Regularization | $1203 \pm 397$ | $0.308 \pm 0.019$ |
| | No Vine and No Regularization | $1237 \pm 78$ | $0.324 \pm 0.057$ |
| Oracle | Oracle Policy | $1474 \pm 205$ | $0.133 \pm 0.034$ |

## 5.1 PERFORMANCE OF TASK POLICY

The training curves for the EPI Policy and three representative baselines are presented in Fig. 2(b) and Fig. 3(b). The final performance of the best Oracle Policy across random seeds is plotted as a dotted line as an upper bound. Average curves are plotted for other policies with standard deviation across seeds. We observe that for both the Hopper and the Striker environments, the EPI Policy achieves better performance than the baselines and is comparable to the performance of the Oracle Policy. To further validate our results, we test our final learned task polices $\pi_{task}$ on heldout test environments and present these results in Table. 1. For Hopper, our method achieves an average episode reward of $1303 \pm 173$, at least $14.0\%$ better than all the baselines. For Striker, our method achieves a final distance of $0.162 \pm 0.015(m)$ to the goal, at least $37.5\%$ more accurate than all the baselines.

The baselines under-perform due to multiple reasons. Without any information of the environment, Simple Policy and Invariant Policy are not able to adapt to the different testing environments. The Random Interaction Policy, the History Policy and the Recurrent Policy, are all able to use environment information and are hence better than the invariant Policy. However, the EPI Policy can learn a lot more efficiently since it has a separate policy that explicitly probes the environment.

Even though a System Identified Policy has extra access to true environment parameters during training time, its performance will be limited when the number of changing parameters increases. To further verify this, we trained System Identified Policy for Hopper when only 2 variables are changing. It achieves a low mean squared error of $10^{-5}$ for environment parameter estimation. However, when 8 parameters of the Hopper are changing, the mean squared error is around $10^{-2}$. Furthermore, this baseline still relies on the assumption that the task-specific policy will capture environment nuances just like the History Policy. This may limit the performance in Striker even when only 2 parameters are changing.

Finally, the Direct Reward Policy learns slowly and is hence worse than using EPI-policies. We attribute this to the added difficulty in credit assignment, i.e. a high reward for the task-specific policy is not necessarily caused by a good interaction, but the interaction policy would still receive the high reward.

## 5.2 ENVIRONMENT EMBEDDING

To understand what the EPI-policy learns, we visualize the embeddings given by interactions. We run interactions over the randomized environments and plot the output embedding in Fig. 2(c) for the Hopper and in Fig. 3(c) for the Striker. The embedding shows a correspondence to environment parameters with similar environments being closer to each other. This means that the agent learns to disentangle state transition differences induced by different environment parameters. This also demonstrates that it has learned to generate distinguishable trajectories in differing environments.

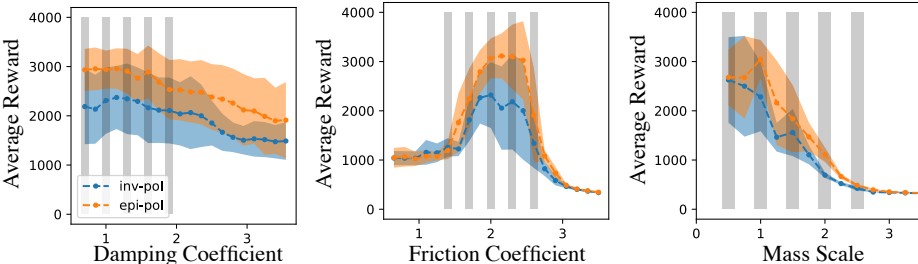

Figure 4: Generalizability of the EPI-policy on Hopper. The grey area represents the training range while the rest represents testing environments.

## 5.3 ANALYSIS ON GENERALIZABILITY

To evaluate the generalizability of the EPI-policy, we further analyze the performance of Hopper when only damping, friction or mass is changing in Fig. 4. During the analysis of one parameter, the other parameters are set to default values. Grey area represents training range, while the rest of the values is only seen in testing range. From the figure we can see that our policy outperforms the invariant policy in almost every setting.

## 5.4 ABLATIVE EXPERIMENTS

**Effect of Vine-Method for Initial Dataset:** Vine method allows the transition prediction model to not overfit on samples based on their state. Training without the Vine method, leads to dropping performance by about 89 reward points on Hopper and 0.13m accuracy on Striker. This is still better than all baselines for Hopper. This shows that the vine method is important for certain environments like Striker, but it is not necessary.

**Effect of Regularization:** The only thing the additional regularization loss does is trying to cluster embeddings from the same environment. It does not serve for the EPI-policy updates as prediction loss does. However, even without separation loss, EPI is significantly better than the Invariant policy baseline by 178 points on Hopper. Similar to Vine method, separation loss is necessary for Striker but not for Hopper to beat baselines. In addition, we did experiments without either Vine-Method or Regularization. Hopper performs 66 points worse than following the full method, while still beating all the baselines. Striker performs 0.162m worse than the full method. Although it still beats 5 of the 7 baselines, this is a significant loss in performance and highlights the importance of using both the Vine data and regularization.

**Experimental Setup:** Our method probes the environment using the EPI-policy followed by solving the task itself. During task-solving, the agent starts from the same initial states as baselines (using reset). While in most cases, this might be a feasible assumption, initial environment probing might not be allowed and resets might not be available in some scenarios. Therefore, we also performed experiments when the EPI-policy probes the environment for the first 10 steps and then the task-policy takes over from where the EPI-policy left. This essentially leads to a larger range of initial states for the task-specific policy than all the baselines. Even in these strict conditions, EPI-policy is quite useful. For Hopper, we achieve a reward of $1193 \pm 304$, outperforming all the baselines as shown in Table. 1. Even in the case of Striker we perform on par with invariant policy and outperform all other baselines.

## 6 CONCLUSION

We have presented an approach to extract environment behavior information by probing it using an EPI-policy. To learn the EPI-policy we use a reward function which prefers trajectories that improve environment transition predictability. We show that such a reward objective enables the agent to perform meaningful interactions and extract implicit information (environment embedding) from the environment. Given unseen test environments on Hopper and Striker, we show that our embedding conditioned task policies improve generalization to unseen test environments.

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

## APPENDIX A    ENVIRONMENT DESCRIPTIONS

We used Hopper and Striker environments from OpenAI Gym (Brockman et al., 2016). We will describe the details of the environments in this section.

**Hopper:** Hopper consists of four body parts and three joints. It has an 3-dimensional action space including motor commands for all the joints. The observation space is 11-dimensional, inluding joint angles, joint velocities, y,z positions of the torso and x,y,z velocities of the torso. The initial positions and velocities are randomized over $\pm 0.005$. The reward function is $r(s, a) = v_x + 1 - 0.001\|a\|^2$. The first term is to encourage the agent to move forward. The second term is a reward for being alive. The third term is to disencourage unnecessary movements. The criteria for being alive is that the height of the torso should be larger than 0.7 and the angle of the torso should be smaller than 0.2 with respect to the horizontal plane. It has a maximum length of 1000 timesteps for each episode. If it falls down and breaks the alive condition, the episode will end immediately.

**Striker:** Striker environment has a 7-dof robot arm and a table. There is a ball and a goal location marker on the table. The action space is 7-dimensional, including the motor commands for each joints on the robot. The 23-dimensional observation space consists of the joint angles, joint velocities, the end-effector (hand) position, the object position and the goal position. The initial conditions randomize the velocity of all the joints over $\pm 0.1$. The reward is given by $r(s, a) = -3\|s_{object} - s_{goal}\| - 0.1\|a\|^2 - 0.5\|s_{object} - s_{hand}\|^2$. The first term is to encourage the object to reach the goal and stay at the goal. The second term is to disencourage unnecessary movements. The third term is to encourage the hand to hit the object. After the end-effector hits the object, $s_{hand}$ will be replaced by the hitting location of the end-effector. It has a fixed length of 200 timesteps for each episode.

## APPENDIX B    RANDOMIZED ENVIRONMENT PARAMETERS

Every environment parameter is randomized among 5 evenly spread discrete values. Striker has 2 varying parameters which create 25 possible environment. Hopper is varying 8 parameters, so there are $5^8$ possible environments. In order to calculate separation loss for Hopper, we sample 100 environments from all $5^8$ possible environment to train the EPI-policy. Otherwise it will not be able to cluster the embeddings from the same environment. Note that when training task policy with interactions, the environment is still randomized from all possible combinations.

During testing, all environment parameters are sampled from a set of discrete values that the policy has never seen in any stage of training. For example, if mass is sampled from $\{1.0, 2.0, 3.0, 4.0, 5.0\}$, it is sampled from $\{1.5, 2.5, 3.5, 4.5, 5.5\}$ during testing.

## APPENDIX C    TRAINING DETAILS

An EPI-trajectory contains 10 steps of observations and actions for both Hopper and Striker. The embedding network $\psi$ that inputs an EPI-trajectory $\tau_{epi}$ and outputs environment embedding $\varepsilon$ has two fully connected layers with 32 neurons each and followed by ReLU (Krizhevsky et al.). The prediction models $f_{pred}$ and $f_{epi\_pred}$ has four fully connected layers with 128 neurons each. The prediction model is optimized by Adam (Kingma & Ba, 2014). Both the EPI-policy $\pi_{epi}$ and the final goal policy $\pi_g$ are optimized using TRPO (Schulman et al., 2015) with rllab implementation (Duan et al., 2016a). The prediction model is trained from scratch every 50 policy updates. Training from scratch is to avoid overfitting which will lead to unintended increasing reward for the EPI-policy. The EPI-policy is trained for 200∼400 iterations in total with a batch size of 10000 timesteps. The task policy will then use the trained EPI-policy and the embedding network to update for 1000 iterations with a batch size of 100000 timesteps.

## APPENDIX D    TRAINING DETAILS OF BASELINES

All MLP baselines have the same network architecture as the task policy: two hidden layers with 32 hidden units each with ReLU activations. They are trained with the same batch size, optimizer,

number of iterations and other hyperparameters as the task policy. We made sure that all training curves saturate before the final iteration.

- Simple Policy: a MLP policy trained only in one fixed environment of fixed parameters. All the other baselines are trained across the randomized environments.

- Invariant Policy (*inv-pol*): a default MLP policy $a_t = \pi(s_t)$.

- Oracle Policy: a MLP policy takes explicit environment parameters as additional input: $a_t = \pi(s_t, \rho)$, where $\rho$ is a vector of the environment parameters such as mass, friction and damping.

- Random Interaction Policy: a MLP policy that takes 10 timesteps of random interactions as additional input. $a_t = \pi(s_t, s_{i0}, a_{i0}, s_{i1}, a_{i1}...s_{i9}, a_{i9})$, where $s_{in}$ and $a_{in}$ are from a random policy.

- History Policy: a MLP policy that takes 10 timesteps of most recent states and actions as additional input. Mnih et al. (2015) adopts similar idea where it stacks 3 to 5 most recent frames as input. $a_t = \pi(s_t, s_{t-1}, a_{t-1}, s_{t-2}, a_{t-2}...s_{t-10}, a_{t-10})$. When $t < 10$, $s_{t-j}$ are filled by zeros for $j > t$.

- Recurrent Policy (*lstm-pol*): The Recurrent baseline is using an LSTM with 32 hidden units from rllab. $a_t = \pi_{LSTM}(s_t)$.

- System Identified Policy: This baseline is implemented based on Yu et al. (2017). We first trains an Oracle Policy $a_t = \pi(s_t, \rho)$ as we described above, which is called the Universal Policy (UP) in their paper. To train Online Sytem Identification (OSI) model, we collect trajectories on randomized environments following the Oracle Policy. With the trajectories, we trained the OSI model, which is a regression model that inputs a history of states and actions and outputs $\rho$. The network architecture strictly follows the paper. Then the estimated $\rho$ are fed into the trained Oracle Policy $a_t = \pi(s_t, \rho)$ to collect more trajectories for training OSI again. We followed the original paper and trained OSI for 5 times.

- Direct Reward Policy: takes the first 10 steps of actions as additional input. In this way, these steps are taking the future reward of the whole trajectory. $a_t = \pi(s_t, s_0, a_0, s_1, a_1...s_9, a_9)$. When $t < 10$, $s_j$ are filled by zeros for $j > t$.

