# OpenReview forum: "Environment Probing Interaction Policies"
_ICLR.cc/2019/Conference_

### Official Review · AnonReviewer3 · 2018-11-01
**Learning policies for probing environment parameters to accelerate task learning. Interesting though limited novelty.**

**Rating:** 6
**Confidence:** 4

**Review:**

The submission presents a reinforcement learning method for exploring/probing the environment to determine an environment’s properties and exploit these during later tasks. The method relies on jointly learning an embedding that simplifies prediction of future states and a policy that maximises a curiosity/intrinsic motivation like reward to learn to explore areas where the prediction model underperforms. In particular, the reward is based on the difference between prediction based on the learned embedding and prediction based on a prior collected dataset, such that the reward optimises to collect data with a large difference between the prediction accuracy of both models. The subsequently frozen policy and embedding are then used in other domains in a system identification like manner with the embedding utilised as input for a final task policy. The method is evaluated on a striker and hopper environment with varying dynamics parameters and shown to outperform a broad set of baselines.

In particular the broad set of baselines and small performed ablation study on the proposed method are quite interesting and beneficial for understanding the approach. However, the ablation study could be in more detail with respect to the additional training variations (Section 4.1.3; e.g. without all training tricks). Additionally, information about the baselines should be extended in the appendix as e.g. different capacities alone could have an impact where the performances of diff. algorithms are comparably similar. In particular, additional information about the training procedure for the UP-OSI (Yu et al 2017) baseline is required as the original approach relies on iterative training and it is unclear if the baseline implementation follows the original implementation (similar to Section 4.1.3.).

Overall the submission provides an interesting new direction on learning system identification approaches, that while quite similar to existing work (Yu et al 2017), provides increased performance on two benchmark tasks. The contribution of the paper focuses on detailed evaluation and, overall, beneficial details of the proposed method. The novelty of the submission is however limited and highly similar to current methods.

Minor issues:
- Related work on learning system identification:
Learning to Perform Physics Experiments via Deep Reinforcement Learning
Misha Denil, Pulkit Agrawal, Tejas D Kulkarni, Tom Erez, Peter Battaglia, Nando de Freitas

---

> ### Author Response · Authors · 2018-11-07
> **Response to AnonReviewer3**
>
> Thank you for your detailed review and for finding our work interesting. We will focus our response on the novelty of our paper and clarify some key contributions.
>
> Novelty in reward formulation compared to intrinsic motivation:
> For an intrinsic motivation/curiosity reward, the reward would be the error in prediction. This would encourage the policy to explore unexplored regions of state space. However, in our work, the reward is the difference in prediction error of two models. The first model predicts directly while the second model is conditioned on EPI-trajectory. This is an important distinction which causes our EPI-policy to extract information to improve prediction and not explore. Using errors in prediction modelling as a surrogate for environment information gathering is to the best of our knowledge novel to our work. We attempted to highlight this difference in the last paragraph of our ‘Related Work’ section. However, we will further highlight this difference in the approach section.
>
> Novelty compared to UP-OSI (Yu et al. 2017):
> UP-OSI is an approach for explicit system identification. It takes in trajectories optimized for a specific task and tries to predict the environment parameters directly. But those trajectory to solve a task may not be optimal to disentangle to effects of different environment parameters (Lowrey et al. 2018). Hence this could make explicit prediction impractical for a large number of entangled environment parameters.
> Instead, our EPI policy is optimized to extract underlying parameters represented via embedding. This also reduces the burden of exactly disentangling the environment parameters, while providing sufficient information to learn a task. Furthermore, our results demonstrate that having a separate policy to extract embedding is a better strategy to get higher rewards.
>
> Ablation studies:
> As per your suggestion, we have run the ablation experiment without using any training tricks (i.e. “No Vine Data” + “No Regularization”). Hopper achieves 1237 pts which is 66 pts worse than using all the tricks. It however still beats all baselines. For the Striker, we get 0.324m final distance which is 0.162m worse than using all the tricks. Although we still beat 5/7 baselines, this is a significant loss in performance and highlights the importance of using both the Vine data and regularization. The ablation results of individual training tricks can be seen in Section 5.4.
>
> Description of baselines:
> For baselines, we will add more information in the Appendix. The MLP policies, baselines and our policy, has two hidden layers with 32 hidden units each with relu activations. This ensures that the capacity of the networks are the same. The Recurrent baseline is using an LSTM with 32 hidden units from rllab. All of these network sizes are defaults from the rllab toolbox. For the UP-OSI baseline, we followed the original paper and ran 5 iterations for training OSI.
>
> Related work:
> We thank the reviewer for this reference and we will include it in the Related Work section.
>
> Thank you again for reviewing our paper. We would be happy to provide any further clarifications. We will upload an updated version of the paper with the appendix and more details in a few days.
>
> References:
> Lowrey, Kendall, et al. "Reinforcement learning for non-prehensile manipulation: Transfer from simulation to physical system." Simulation, Modeling, and Programming for Autonomous Robots (SIMPAR), 2018 IEEE International Conference on. IEEE, 2018.

---

> > ### Comment · AnonReviewer3 · 2018-11-21
> > **Response**
> >
> > Thanks for the feedback. The changes have rendered the contributions more clear and in particular the ablation studies are appreciated with respect to clarification of individual contributions of framework aspects. The previous rating stays appropriate due to similarity to existing approaches and ablated performance results.

---

### Official Review · AnonReviewer2 · 2018-11-02
**Interesting work about environment generalization for reinforcement learning**

**Rating:** 6
**Confidence:** 2

**Review:**

This paper proposes an “Environment-Probing” Interaction (EPI) policy used as an additional input for reinforcement learning (RL). This EPI allows to extract environment representations and implicitly understand the environment in order to improve the generalization on novel testing environments.

Pros:

This paper is well written and clear and the contribution is relevant to ICLR. Although I am not familiar with RL , the contribution seems novel and the model performances are compared with strong and appropriate baselines.

---

> ### Author Response · Authors · 2018-11-07
> **Response to AnonReviewer2**
>
> Thank you for your review and for finding our paper well-written and clear. We are happy to address any other concerns or questions about our work.

---

### Official Review · AnonReviewer1 · 2018-11-05
**great paradigm, but paper could be more efficient**

**Rating:** 6
**Confidence:** 3

**Review:**

Some argumentation might better be supported by some reference, like :

"When humans are tasked to perform in a new environment, we do not explicitly know what param-
eters affect performance. Instead, we probe the environment to gain an intuitive understanding of
its behavior (Fig. 1). The purpose of these initial interactions is not to complete the task imme-
diately, but to extract information about the environment. This process facilitates learning in that
environment. Inspired by this observation,
"

The overall idea is interesting, the implementation is correct via a TRANSITION PREDICTION MODELS

More place could be taken for more detailed results, use appendix to swap some text...

---

> ### Author Response · Authors · 2018-11-07
> **Response to AnonReviewer1**
>
> Thank you for your review and for finding our work interesting. We will add appropriate references at places in introduction. For example, Braun et al. show that humans indeed adapt online within a single trial to perform a task in unpredictable environments rather than using a fixed policy. We will add similar references to the paper.
>
> We are also working on expanding the discussion to give more details on the results. We will upload a new version in few days. If you have any specific suggestions or additional questions, we will be happy to address them.
>
> Reference:
> Daniel A Braun, Ad Aertsen, Daniel M Wolpert, and Carsten Mehring. Learning optimal adaptation strategies in unpredictable motor tasks. Journal of Neuroscience, 29(20), pp.6472-6478, 2009.

---

### Author Response · Authors · 2018-11-15
**New version**

We thank all the reviewers for reviewing our paper. We have updated the paper with the following changes according to the suggestions:

- Added a reference in the introduction section (R1)
- Added a reference in the related work section (R3)
- Highlighted the difference of our reward comparing to the curiosity reward in the approach section 4.1.2 (R3)
- Additional experiment results for ablation studies in Table 1 and additional discussion in section 5.4. (R1, R3)
- Provided more details of the simulation environments in Appendix A
- Provided more implementation details of the baselines in Appendix D (R3)
- Minor changes to fit in 8 pages

---

### Meta-Review · Area_Chair1 · 2018-12-15

**Confidence:** 4
**Recommendation:** Accept (Poster)

**Metareview:**

This paper proposes an approach for probing an environment to quickly identify the dynamics. The problem is relevant to the ICLR community. The paper is well-written, and provides a detailed empirical evaluation. The main weakness of the paper is the somewhat small originality over prior methods on online system identification. Despite this, the reviewer's agreed that the paper exceeds the bar for publication at ICLR. Hence, I recommend accept.

Beyond the related work mentioned by the reviewers, the approach is similar to work in meta-learning. Meta-RL and multi-task learning has typically been considered in settings where the reward is changing (e.g. see [1],[2],[3],[4], where [4] also uses an embedding-based approach). However, there is some more recent work on meta-RL across varying dynamics, e.g. see [5],[6]. The authors are encouraged to make a conceptual connection between this approach and the line of work in model-based meta-RL (particularly [5] and [6]) in the final version of the paper.

[1] Duan et al. https://arxiv.org/abs/1611.02779
[2] Wang et al. CogSci '17 https://arxiv.org/abs/1611.05763
[3] Finn et al. ICML '17 https://arxiv.org/abs/1703.03400
[4] Hausman et al. ICLR '17: https://openreview.net/forum?id=rk07ZXZRb
[5] Sæmundsson et al. https://arxiv.org/abs/1803.07551
[6] Nagabandi et al. https://arxiv.org/abs/1803.11347